# Research

evolution

symbiosis, mitochondria, population genetics

**Author for correspondence:**
Gregory D. D. Hurst
e-mail: g.hurst@liv.ac.uk

# Positive selection on mitochondria may eliminate heritable microbes from arthropod populations

Andy Fenton[1], M. Florencia Camus[2] and Gregory D. D. Hurst[1]

[1]Institute of Infection, Veterinary and Ecological Sciences, University of Liverpool, Liverpool, UK
[2]Research Department of Genetics, Evolution and Environment, University College London, Gower Street, London WC1E 6BT, UK

AF, 0000-0002-7676-917X; MFC, 0000-0003-0626-6865; GDDH, 0000-0002-7163-7784

Diverse eukaryotic taxa carry facultative heritable symbionts, microbes that are passed from mother to offspring. These symbionts are coinherited with mitochondria, and selection favouring either new symbionts, or new symbiont variants, is known to drive loss of mitochondrial diversity as a correlated response. More recently, evidence has accumulated of episodic directional selection on mitochondria, but with currently unknown consequences for symbiont evolution. We therefore employed a population genetic mean field framework to model the impact of selection on mitochondrial DNA (mtDNA) upon symbiont frequency for three generic scenarios of host–symbiont interaction. Our models predict that direct selection on mtDNA can drive symbionts out of the population where a positively selected mtDNA mutation occurs initially in an individual that is uninfected with the symbiont, and the symbiont is initially at low frequency. When, by contrast, the positively selected mtDNA mutation occurs in a symbiont-infected individual, the mutation becomes fixed and in doing so removes symbiont variation from the population. We conclude that the molecular evolution of symbionts and mitochondria, which has previously been viewed from a perspective of selection on symbionts driving the evolution of a neutral mtDNA marker, should be reappraised in the light of positive selection on mtDNA.

## 1. Introduction

Maternally inherited microbes—bacteria, viruses and fungi that pass from a female to her progeny—are found commonly in eukaryotic hosts. Most commonly recognized as bacterial associates of arthropods, like *Wolbachia* [1], heritable microbes are also found in plants [2,3], diverse microeukaryotes (e.g. [4]) and a range of invertebrates (e.g. [5,6]). These microbes modify the biology of their host individual, able to act either as beneficial symbionts that contribute to organismal function or as reproductive parasites, and interactions may be obligate (the host requires the symbiont, and typically all individuals are infected) or facultative (the host remains able to survive and reproduce without the symbiont, commonly with a mix of symbiont-infected and -uninfected host individuals). Contributions to function commonly include the synthesis of amino acids and essential cofactors and the protection of the host against attack by natural enemies (virus, parasitic wasps, nematode worms, fungi and predators) [7]. Reproductive parasitism is a consequence of the exclusive maternal inheritance of the microbe, which selects for strains that bias investment towards the production and survival of female offspring over male and for incompatibility between infected males and uninfected females [8]. These individual impacts affect population and community dynamics and are additionally important drivers of host evolution.

Heritable microbes are maternally inherited alongside mitochondria, and it has long been recognized that the spread of heritable microbes leads to the spread of the mitotype originally associated with the symbiont [9]. Thus, mitochondrial evolution and diversity is a product in part of the recency with which either a new symbiont, or a mutation of an existing symbiont strain, has spread into the population [10]. This process is sufficiently strong that symbionts may also drive mitochondrial introgression across species boundaries following rare hybridization events [11], and as such represent potential disruptors of the 'barcoding gap' [12].

More recently, it has become clear that mitochondria are not the neutral marker they have been historically considered but are, in fact, themselves subject to direct selection [12–14]. Direct selection may be associated with their vital role in energy and heat production, as well as dietary and thermal adaptation [15,16]. Mitonuclear coadaptation is also recognized, in which selection acts jointly on mitochondrial and nuclear variants to maintain a functional integrated mitonuclear system [17,18]. It is thus clear that the current view—that mitochondrial diversity is shaped by symbionts—is likely to represent one component of a reciprocal interaction. We would additionally expect selection on mitochondria to impact on the dynamics and diversity of heritable microbes. If this is true, the outcomes of mitochondrial selection may vary depending on the phenotype of, and direct selection pressures experienced by, the symbiont. However, these possibilities have yet to be formally explored.

In this paper, we use mathematical models to explore the impact of positive selection on mitochondria upon the dynamics of facultative heritable symbionts. We consider facultative microbes because, unlike obligate heritable microbial symbionts (those required for host function), facultative heritable microbes are commonly present in a fraction of host individuals, with a population carrying a mix of symbiont-infected and symbiont-uninfected individuals. Thus, a novel beneficial mitotype can arise initially on either a symbiont-infected background, or on a symbiont-uninfected background. When the novel mitotype arises, it will alter the relative fitness of the two different backgrounds and thus the frequency with which these occur, with potential implications for equilibrium frequency and persistence of the symbiont.

We perform this analysis for three dynamical categories of heritable microbe. First, we consider symbionts maintained by a balance of a direct drive phenotype that is fixed in magnitude, alongside segregational loss. The drive in this model class approximates to that obtained through resource reallocation associated with reproductive parasitic phenotypes like male-killing [19], as well as host beneficial symbiont traits such as nutritional benefit. We then consider symbionts maintained by a benefit that declines as they become more common (negative frequency dependence). This class of models is applicable to defensive symbionts where symbiont spread is driven by imparting resistance to natural enemies, but as the symbiont becomes more frequent, so natural enemy attack rates (and the benefits of symbiont carriage) may decline [20]. Finally, we explore symbionts where the symbiont drive is positively frequency-dependent, that is to say the advantage of having the symbiont becomes stronger when the symbiont is more common. For this class, we model the particular case of cytoplasmic incompatibility

(CI), where the uninfected zygotes die if the fertilizing sperm comes from an infected male [21].

## (a) Modelling frameworks

To understand the coevolutionary dynamics of a potential new mitotype and a symbiont, we initially assume the symbiont is at equilibrium with the resident mitotype and seek the criteria which lead to invasion (and possible replacement of the resident mitotype) by a new mitotype, alongside the consequences for the persistence and equilibrium prevalence of the symbiont. For all scenarios considered (see below), we model the change in frequencies of individuals that carry the resident mitotype and the symbiont ($p$), those that carry the resident mitotype but not the symbiont ($q$), those that carry the new mitotype and the symbiont ($r$), and those carrying the new mitotype but not the symbiont ($s$). We initially assume the symbiont is at equilibrium with the resident mitotype (i.e. the system is at $\{p = p_R^*, q = q_R^*, r = 0, s = 0\}$, where $p_R^*$ and $q_R^*$ are the relevant equilibrium frequencies in the presence of the resident mitotype alone). A new mitotype then emerges that confers a fitness benefit of $t$ to all individuals that carry it. In what follows we mostly focus on values up to $t = 0.02$, reflecting strong selection on the mitochondrial DNA (mtDNA) (a 2% benefit to the novel mitotype), although in some cases we present results for a wider range of values for completeness.

The three frameworks for symbiont dynamics are modelled generically as a fixed benefit, a negative frequency-dependent benefit, or CI with a positive frequency benefit. For ease, these formulations are simplified from real-world biological scenarios in some cases. For instance, in reality, the fixed benefit to male-killing passes both to infected females and any uninfected siblings generated through segregational loss [19], whereas in our model the transfer to the low number of uninfected siblings is ignored for ease. Likewise, a real-world case of negative frequency-dependent selection would likely include an account of eco-evolutionary dynamics with host and natural enemy density changing dynamically [20]). Thus, our models approximate real-world scenarios without precisely mirroring them. In our account below, we provide the recursion equations for each model; other mathematical details can be found in the electronic supplementary material.

### (i) Model i: facultative heritable symbionts with a fixed benefit

For this scenario, we assume a weakly beneficial symbiont, such that offspring of infected mothers receive a net fitness benefit $(1 + B)$; this benefit is a composite of the fixed benefit supplied and the fixed costs of symbiont carriage. The symbiont is assumed to have vertical transmission efficiency $(1 - \mu)$, such that a proportion $\mu$ of offspring born to infected mothers do not carry the symbiont (segregational loss). As stated above, the new mitotype is assumed to confer a selective benefit $t$ to individuals that carry it. The relative numbers of viable offspring born to each maternal type are given in electronic supplementary material, table S1, and the changes in frequencies of each demographic group from generation $T$ to $T + 1$ are then

given by the recursion equations:

$$p_{T+1} = \frac{p_T(1-\mu)(1+B)}{p_T(1+B)+q_T+(r_T(1+B)+s_T)(1+t)},$$

$$q_{T+1} = \frac{p_T(1+B)\mu+q_T}{p_T(1+B)+q_T+(r_T(1+B)+s_T)(1+t)},$$

$$r_{T+1} = \frac{r_T(1+B)(1-\mu)(1+t)}{p_T(1+B)+q_T+(r_T(1+B)+s_T)(1+t)}$$

and

$$s_{T+1} = \frac{(r_T(1+B)\mu+s_T)(1+t)}{p_T(1+B)+q_T+(r_T(1+B)+s_T)(1+t)}.$$

Here, we establish the conditions for the symbiont to persist in the presence of the resident mitotype alone by setting $r = s = 0$, resulting in the system being described by the single equation (since $q = 1 - p$):

$$p_{T+1} = \frac{p_T(1-\mu)(1+B)}{1+Bp_T}.$$

There are two equilibria to this system: $p^* = 0$ and $p^* = 1 - ((1+B)\mu/B)$; the latter of which is stable if $\mu < B/(1+B)$, the criterion for symbiont persistence. In the electronic supplementary material, we establish the stability criteria of the full system to determine under what conditions the new mitotype can invade and the consequences for the symbiont.

### (ii) Model ii: facultative heritable symbionts whose benefit declines as they become more common

Next we model a symbiont whose benefit declines with its frequency, and for simplicity, we assume a linear relationship between the benefit $B$ and overall symbiont frequency $(p+r)$ of the form $B_T = b(1-(p_T+r_T))$, where $b$ is the maximum benefit when the symbiont is vanishingly rare. We also include a fixed cost, $c$, to symbiont carriage, for example through a reduction in host fecundity. As before we assume a rate of segregational loss of the symbiont, $\mu$. The numbers of viable offspring born to each maternal type are given in electronic supplementary material, table S2, and the changes in frequencies of each demographic group from generation $T$ to $T+1$ are then given by the recursion equations:

$$p_{T+1} = \frac{p_T(1-\mu)(1+B_T-c)}{p_T(1+B_T-c)+q_T+(r_T(1+B_T-c)+s_T)(1+t)},$$

$$q_{T+1} = \frac{p_T(1+B_T-c)\mu+q_T}{p_T(1+B_T-c)+q_T+(r_T(1+B_T-c)+s_T)(1+t)},$$

$$r_{T+1} = \frac{r_T(1+B_T-c)(1-\mu)(1+t)}{p_T(1+B_T-c)+q_T+(r_T(1+B_T-c)+s_T)(1+t)}$$

and

$$s_{T+1} = \frac{(r_T(1+B_T-c)\mu+s_T)(1+t)}{p_T(1+B_T-c)+q_T+(r_T(1+B_T-c)+s_T)(1+t)}.$$

As before we establish the conditions for symbiont persistence in the presence of the resident mitotype alone ($r = s = 0$). Here the system can be described by the equation:

$$p_{T+1} = \frac{p_T(1-\mu)(1+b(1-p_T)-c)}{p_T(1+b(1-p_T)-c)+(1-p_T)}.$$

There are two equilibria to this system: $p^* = 0$ and $p^* = (b(2-\mu)-c-\sqrt{c^2+b\mu(4+b\mu-2c)})/2b$, which is stable if $\mu < (b-c/1+b-c)$, the criterion for symbiont persistence. Stability analysis of the full system is again presented in the electronic supplementary material.

### (iii) Model iii: facultative heritable symbionts whose benefit increases as they become more common

Finally we consider a symbiont under positive frequency-dependent selection, like one that confers CI. In such a model, it is assumed that crosses between uninfected females and infected males reduce offspring fitness by a factor $(1-h)$. As before we also assume a rate of segregational loss of the symbiont, $\mu$. For simplicity we ignore potential costs of symbiont carriage, as at equilibrium these are vastly outweighed by benefits under positive frequency dependence.

Because we have to consider the effect of different male–female mating combinations to account for CI, we now explicitly keep track of the frequencies of uninfected and infected males and females of each mitotype. The numbers of viable offspring born to each maternal–paternal combination are given in electronic supplementary material, table S3, and the overall changes in frequencies of each group from generation $T$ to $T+1$ are:

$$p_{T+1} = \frac{p_T(1-\mu)}{\Phi},$$

$$q_{T+1} = \frac{(p_T\mu+q_T)((p_T+r_T)(1-h)+q_T+s_T)}{\Phi},$$

$$r_{T+1} = \frac{r_T(1-\mu)(1+t)}{\Phi}$$

and

$$s_{T+1} = \frac{(r_T\mu+s_T)(1+t)((p_T+r_T)(1-h)+q_T+s_T)}{\Phi},$$

where $\Phi$ is the sum of all the offspring values in electronic supplementary material, table S3, weighted by their corresponding mating frequency.

Again the conditions for symbiont persistent in the presence of the resident mitotype alone can be found by setting $r = s = 0$, reducing the system to the single equation:

$$p_{T+1} = \frac{p_T(1-\mu)}{1-hp_T(1-p_T(1-\mu))}.$$

There are three equilibria to this system:

$E_1: p^* = 0,$

$E_2: p^* = \dfrac{h-\sqrt{h^2-4h\mu(1-\mu)}}{2h(1-\mu)}$

and $E_3: p^* = \dfrac{h+\sqrt{h^2-4h\mu(1-\mu)}}{2h(1-\mu)},$

$E_3$ is the positive internal equilibrium for the symbiont, which is biologically feasible if $\mu < \left(1-\sqrt{1-h}\right)/2$. However, due to the positive frequency dependency inherent in the CI system, this criterion is not sufficient to guarantee symbiont persistence (e.g. [22]). State $E_2$ is an unstable equilibrium, such that the initial frequency of the symbiont, $p_0$, has to exceed that the equilibrium value in order to reach the internal equilibrium $E_3$; if not then the system heads to $E_1$, and the symbiont fails to persist. Unfortunately, a mathematical analysis of the full system involving the invading mitotype is not possible—for that reason we resort to analysis based on numerical simulations.

For each scenario, we analyse separately the dynamics arising when the novel mitotype emerges either in a symbiont-infected individual or one without the symbiont. We then explore the stability criteria to establish the criteria for this invading mitotype to establish, and the consequences for persistence and prevalence of the symbiont. All simulations were carried out using Mathematica

v.12.1 (code available at https://doi.org/10.6084/m9.fig-share.c.5354999.v2).

## 2. Results

### (a) Facultative heritable symbionts with a fixed benefit

In our mean field model, the symbiont is present in the population if the symbiont's rate of segregational loss, $\mu$, is less than $B/(1+B)$, where $B$ is the net benefit of symbiont infection (see electronic supplementary material, §i). Positively selected mitochondria ($t > 0$) that arise in a symbiont-infected individual spread to fixation. Symbiont prevalence increases during spread, as the advantageous mitotype is initially more common within symbiont-infected individuals than uninfected ones. However, the prevalence of the symbiont at equilibrium is not altered, as the beneficial mitotype later establishes into the uninfected population following symbiont segregational loss (figure 1). This process is easily visualized as generating a novel cytotype (advantageous mitotype + symbiont) that has higher benefit than all other cytotypes (advantageous mitotype—no symbiont; ancestral mitotype + symbiont; ancestral mitotype—no symbiont). The spread of the advantageous mitotype displaces other symbiont strains from the population (no additional mtDNA benefit), and segregational loss of the symbiont results in the mtDNA also invading through the uninfected portion of the population.

A different pattern is observed when the novel beneficial mitotype mutation arises originally in an uninfected individual (figure 2; see electronic supplementary material, §i, for mathematical details). When a threshold benefit to the novel mitotype of $t$ is reached (above the diagonal line given by $t = B(1 - \mu) - \mu$, the novel mitotype-infected females leave a greater number of progeny than the symbiont-infected females leave symbiont-infected progeny, and the novel mitotype invades and displaces the heritable microbe from the population. If this threshold is not reached ($t < B(1 - \mu) - \mu$, below the diagonal line in figure 2), then the novel mitotype does not invade if it arises in an uninfected individual, despite being advantageous compared with the ancestor. For realistic values of mitotype advantage ($t = 0.02$, or a 2% benefit to the novel mitotype), exclusion is confined to symbionts that exist at low prevalence—where fewer than 20% of hosts carry the symbiont (these symbionts either have low benefit or a higher benefit with high rates of segregational loss).

### (b) Facultative heritable symbionts whose benefit declines as they become more common

We modelled the dynamics of mtDNA for a symbiont whose benefit declined with its overall frequency in the population, for example to mimic a defensive symbiont. For these symbionts, the frequency-dependent benefit exists alongside a fixed cost of carrying the symbiont $c$, such that there is a polymorphic equilibrium for symbiont infection, at a frequency where the benefit of the phenotype is counterbalanced by the cost of infection and any segregational loss $p^* = (b(2 - \mu) - c - \sqrt{c^2 + b\mu(4 + b\mu - 2c)})/2b$. Part of our analysis focused in detail on the special case where a symbiont-infected/-uninfected polymorphism is retained without segregation ($\mu = 0$) (a unique feature of negative

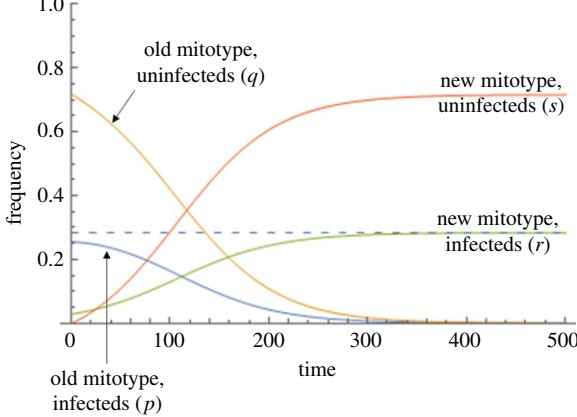

**Figure 1.** Dynamics of novel mitochondria that confer a selective benefit $t$ to individuals that carry it, arising in a symbiont-infected individual in a population initially at equilibrium with that symbiont, for the scenario where the symbiont confers a direct benefit, $B$, to its host (Model i). The $y$-axis represents the proportion of population carrying the four different cytoplasmic combinations, and the $x$-axis time in generations. Blue line— the frequency of symbiont with the ancestral mitotype; orange line—the frequency of uninfected females with the ancestral mitotype; green line—the frequency of symbiont-infected individual with the novel beneficial mitotype; red line—the frequency of uninfected females with the novel beneficial mitotype. Parameter values: $t$, the selective benefit of the novel mitochondria $= 0.02$; $B$, the benefit of symbiont infection $= 0.075$; $\mu$, the rate of segregational loss (i.e. the proportion of offspring born to infected mothers do not carry the symbiont) $= 0.05$. The dashed line represents the baseline equilibrium frequency of the symbiont in the presence of resident mitotype alone. (Online version in colour.)

frequency-dependent advantage models for mean field formulations; for all other classes of mean field model, net beneficial symbionts fix if $\mu = 0$).

A novel advantageous mitotype arising in an infected individual will spread through to fixation and, if there is any level of symbiont segregational loss, symbiont equilibrium prevalence will be unaltered as segregational loss results in the beneficial mitotype flowing into the uninfected fraction of the population (see electronic supplementary material, §ii). This process will also lead to the loss of symbiont diversity.

For the special case where vertical transmission of the symbiont is perfect (i.e. all progeny of an infected female carry the symbiont, $\mu = 0$), a novel advantageous mitotype in the infected portion of the population invades that section of the population but does not pass across to the uninfected portion by segregation. This portion thus retains the ancestral mitotype. If the mitotype is sufficiently advantageous to overcome fixed costs of symbiont infection ($t > (b - c)$), the symbiont becomes beneficial at all frequencies, at which point the symbiont will fix. Otherwise, the symbiont-infected portion increases in frequency associated with the additional benefit of carrying the new mitotype, and the ancestral mitotype persists in the symbiont-uninfected fraction of the population, which declines in frequency. Mitotype variation, therefore, becomes partitioned between symbiont-bearing and symbiont-uninfected pools, with the beneficial mitotype present only in symbiont-infected individuals.

We then examined the dynamics of a novel advantageous mitotype arising in the symbiont-uninfected portion of the population (figure 3). The conditions for the invasion of

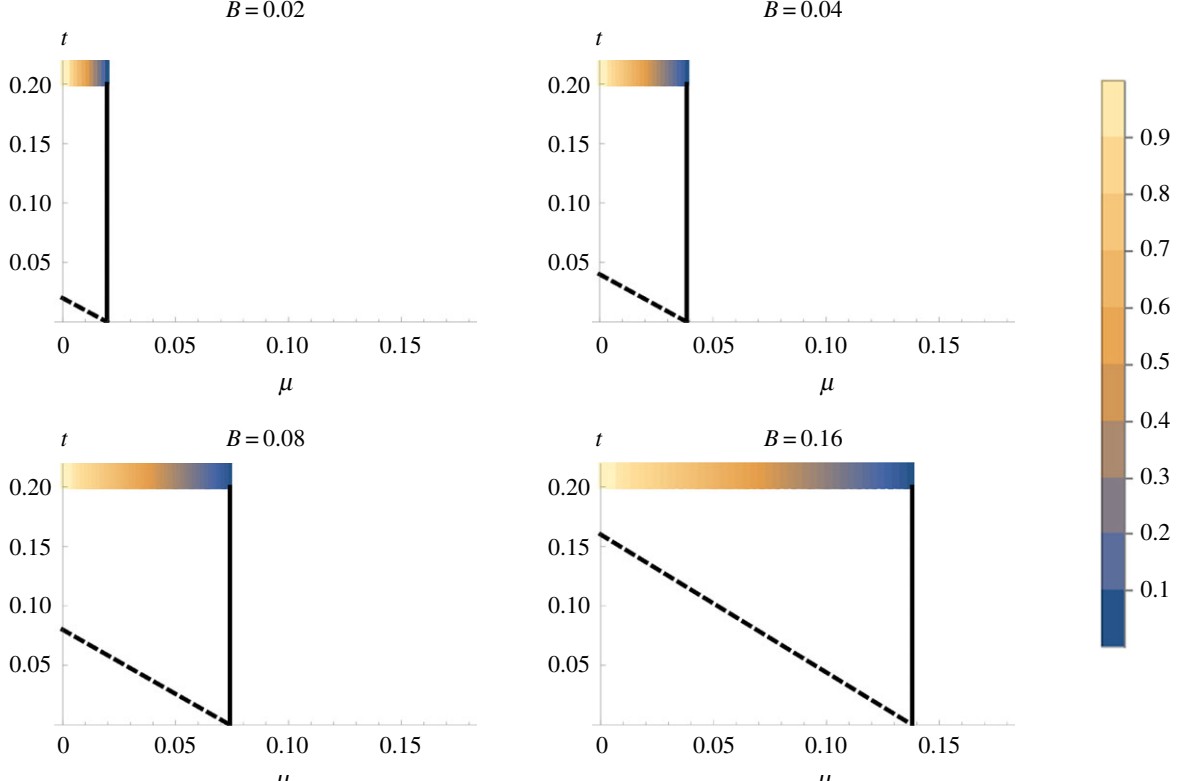

**Figure 2.** Conditions for the invasion of a novel beneficial mitotype arising in a symbiont-uninfected individual where the symbiont has direct benefit, $B$, to its host (Model i). The $x$-axis represents the rate of segregational loss ($\mu$), and the $y$-axis represents the selective benefit of the novel mitochondria ($t$). The heatmap shows the frequency of the symbiont at equilibrium, before mitotype invasion, in each case. Spread and fixation of the novel mitotype occurs above the dashed diagonal line; in this region, the novel mitotype drives the symbiont extinct. The symbiont cannot persist to the right of the solid vertical line given by $\mu = B/(1 + B)$.

this mitotype are less restrictive than for a fixed benefit symbiont, as the negative frequency dependence of the benefit to carrying a symbiont makes the symbiont-infected type only weakly beneficial at equilibrium before the novel mtDNA type arises. At the point of invasion, the symbiont is maintained polymorphic by a benefit–segregational loss balance, and mitotype invasion thus occurs when the benefit of the novel mitotype overcomes the rate of segregational loss from the symbiont-infected section at equilibrium (above the lower curve in figure 3). The invasion of this mitotype lowers symbiont prevalence. The ultimate fate of the mitotype then depends on its level of benefit, $t$. Above the diagonal line in figure 3 (i.e. $t > (1 + b - c)(1 - \mu) - 1$; see electronic supplementary material, §ii), the benefit of the novel mitotype exceeds that of the symbiont at low symbiont frequency, and the symbiont is displaced. In the area between the two lines, the novel mitotype invades the population but does not displace the symbiont. Here, females carrying the novel mitotype leave more progeny than symbiont-infected females when the symbiont is initially at equilibrium, allowing it to invade, but this fitness gap narrows as the symbiont declines in frequency and its benefit increases. These conditions create an equilibrium at which the symbiont-infected type has the ancestral mitotype, and the symbiont-uninfected portion carries the novel mitotype (though not at fixation in this class, due to segregational loss of the symbiont moving the mitotype from the symbiont-infected compartment across to the uninfected compartment) (figure 4).

For the special case of no segregational loss ($\mu = 0$), a novel advantageous mitotype that arises in the uninfected portion of the population rises in frequency and fixes within that section of the population. The symbiont is either depressed in frequency (if $t < b - c$) or is driven from the population (if $t > b - c$).

## (c) Facultative heritable symbionts whose benefit increases as they become more common

We modelled the fate of a novel advantageous mitotype in a population carrying a symbiont maintained by CI. CI is a positively frequency-dependent trait, in which the zygotes produced by infected fathers and uninfected mothers die at a rate $h$, such that a fraction $(1 - h)$ survives. As the frequency of infection in males increases, so does the rate at which uninfected cytotypes become inviable. We examined the fate of mitochondrial mutants that arose in a population at the upper (stable) equilibrium for the symbiont (see electronic supplementary material, §iii).

Taking $t = 0.02$ as the upper limit of possible levels of benefit to a mitochondrial mutation, it was unlikely for a novel mitotype arising in an uninfected individual to invade the population for any level of CI above $h = 0.05$, which is very weak CI (figure 5). Beneficial mitochondrial mutations arising in the symbiont-infected lineage spread to fixation (providing the benefit $t > 0$), moving into the uninfected section of the population following segregational loss of the symbiont (figure 6).

## 3. Discussion

Heritable microbial symbionts and mitochondria are both inherited through the female line, connecting the dynamics

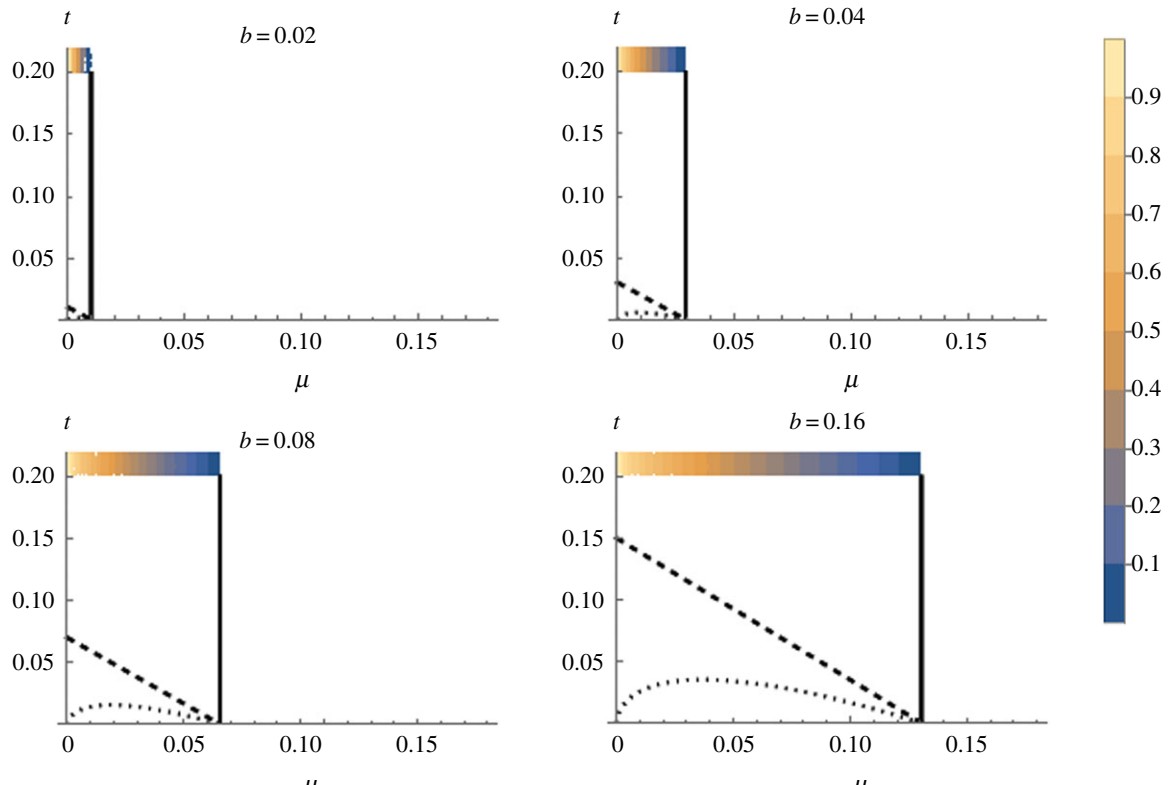

**Figure 3.** Conditions for the invasion of a novel beneficial mitotype arising in a symbiont-uninfected individual where the symbiont has direct benefit to its host, $b$, that declines with the frequency of the symbiont (Model ii). The $x$-axis is the rate of segregational loss ($\mu$), and the $y$-axis represents the selective benefit of the novel mitochondria ($t$). The heatmap is the frequency of the symbiont at equilibrium, before mitotype invasion, in each case. Spread and fixation of the novel mitotype occurs above the dashed diagonal line; beneath this line but above the dotted curve is an area where the mitotype invades but does not fix, remaining polymorphic within uninfected individuals. The symbiont cannot persist to the right of the solid vertical line given by $\mu = (b - c)/(1 + b - c)$. The cost of symbiont carriage $c = 0.01$. (Online version in colour.)

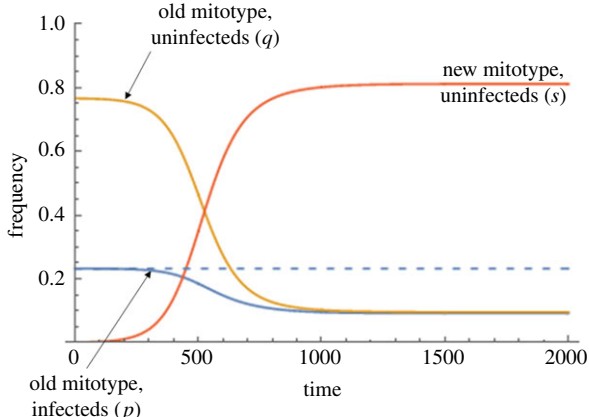

**Figure 4.** Dynamics of a beneficial mitochondrial type ($t = 0.02$) arising in an uninfected individual for a symbiont with the frequency-dependent benefit (Model ii; benefit to the host of symbiont infection, $b = 0.1$, cost to the host of carrying the symbiont, $c = 0.05$, rate of segregational loss of the symbiont, $\mu = 0.02$). The $x$-axis represents time in generations; $y$-axis the frequency of each of the four cytotypes. Blue line—the frequency of symbiont with the ancestral mitochdondrial type; red line—the frequency of uninfected individuals with the new mitotype; orange—the frequency of uninfected individuals with the ancestral mitotype. The dashed line represents the baseline equilibrium frequency of the symbiont in the presence of resident mitotype alone. (Online version in colour.)

of the two elements. The relationship between microbial symbionts and the mitochondria with which they are coinherited has been well explored in terms of the impact of selection on

symbionts driving the evolution of mitochondria [9,23]. Multiple studies have observed symbiont spread driving up the frequency of the associated mitotype. These effects are sufficiently strong that heritable symbionts can drive mtDNA across species boundaries following rare hybridization events [11,24]. Historically, evolutionary biologists have viewed mtDNA as a neutral marker, useful for reflecting patterns of population demography and gene flow. However, both molecular evolution analyses and directed study of phenotype indicate that mtDNA commonly undergoes positive selection [12–14]. These data imply that selection on mtDNA may impact symbiont dynamics, which we tested here. Our analyses centre on the impact of novel, beneficial mutant mitotypes under directional selection on the dynamics of an existing facultative heritable symbiont.

Our analyses indicate that positive selection on mitochondria may drive symbionts extinct when the mutation creating a beneficial mitotype arises in an individual that does not carry the symbiont. The likelihood of this occurring is highest where the symbiont is at low frequency in the population. There are two reasons for such deterministic extinction being most commonly seen in low prevalence infection, besides simple stochastic fadeout at low frequencies. First, positive selection on mtDNA types will only exclude a symbiont where the mutation occurs initially in an uninfected individual, a situation that the balance of numbers makes most likely for low prevalence symbiont infections. Second, symbiont frequency is determined by a balance of drive (the increase in the number of surviving daughters produced

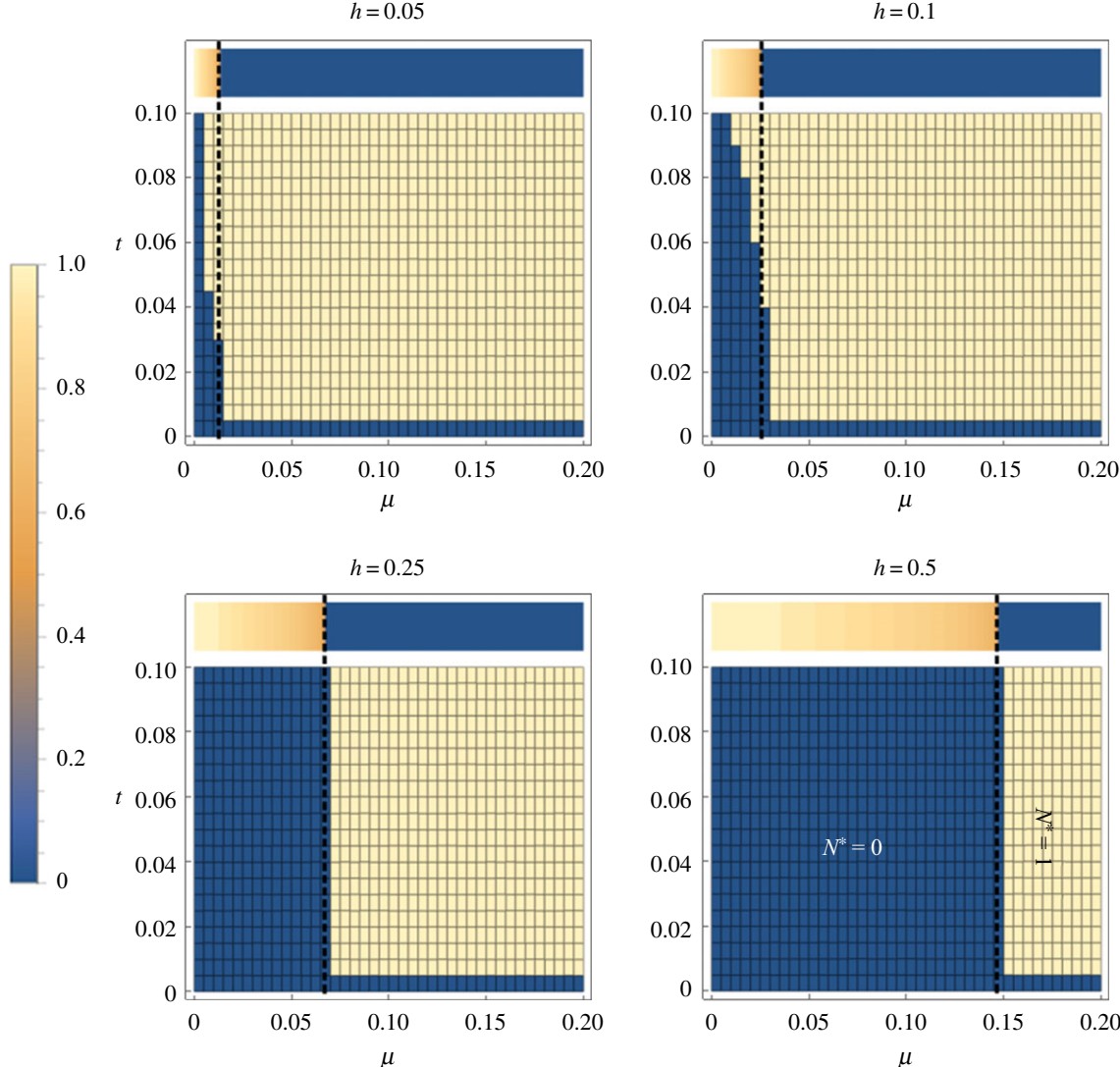

**Figure 5.** Conditions for the invasion of a beneficial mitochondrial mutant arising into a symbiont-uninfected individual for different strengths of CI, $h$ (Model iii). The colours in the plots, relating to the scale on the left, show the overall equilibrium prevalence of the invading mitotype ($N^* = r^* + s^*$; yellow = fixation, blue = absent); the new mitotype invades and replaces the resident in the parameter space in yellow. The $x$-axis is the rate of segregational loss ($\mu$), and the $y$-axis the advantage of the novel mitotype ($t$). The heat bar across the top of each plot is the frequency of the CI symbiont initially before mitotype invasion (again, yellow = fixation, blue = absent). The symbiont cannot persist to the right of the dashed vertical line given by $\mu < \left(1 - \sqrt{1-h}\right)/2$. (Online version in colour.)

by an infected female compared with an uninfected female) and segregational loss (failure of progeny to inherit). The novel mitotype invades when carriers of the novel beneficial mitotype, on average, leave more infected daughters than left by the symbiont-infected ones (carrying the ancestral mtDNA). For selective coefficients on the mtDNA of up to 2%, this process only occurs for low prevalence symbionts.

Low prevalence symbiont infections are thought to be common. Historical studies of well-understood systems where the symbiont has a strong phenotype, such as the *Adalia bipunctata–Rickettsia* interaction, *Drosophila bifasciata–Wolbachia* and *Drosophila–Spiroplasma poulsonii* interaction, indicate that 7, 6, and 1–3% of individuals are infected in these species, respectively [25–27]. PCR screens also reveal that low prevalence infections are common [28], and statistical models of screen data predict that a majority of heritable symbiont infections occur rarely within a species [29]. While the data in screen studies are less robust (a low false positive rate in the PCR assay likely inflates estimates of low prevalence infections), it is nevertheless clear that low prevalence infections are

common. Combining the commonness with which mtDNA is not a neutral marker [12–14] with the high frequency of low prevalence symbiont infections leads us to predict that selection for novel mitotypes will be an important contributor to symbiont turnover.

The lack of shared facultative heritable symbionts in sibling species pairs (e.g. [30]) and indeed between different populations of the same species [1] clearly indicates that loss of facultative heritable microbes from a population occurs commonly. However, the processes that produce symbiont loss are poorly understood. Mitochondrial evolution represents a novel potential driver of this process and is conceptually equivalent to observations of one symbiont replacing another, a process that has been observed in natural populations [31]. The mitochondrial displacement process we have described will act alongside other factors, such as loss of the drive phenotype through suppression by the host [32], loss of the advantage to the phenotype due to ecological change (e.g. loss of a natural enemy for a protective symbiont) or mutational degradation of redundant symbiont phenotype [33].

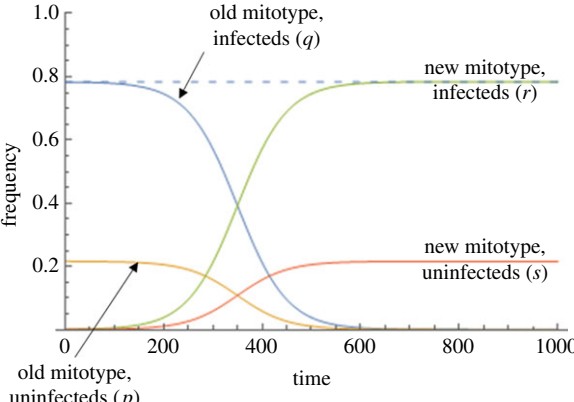

**Figure 6.** Dynamics of a beneficial mitochondrial type ($t = 0.02$) arising in an individual infected by a symbiont that confers CI (Model iii; strength of CI, $h = 0.25$, rate of segregational loss of the symbiont, $\mu = 0.05$). The x-axis is time in generations; the y-axis the frequency of the four different cytoplasmic states. Blue line—the frequency of symbiont with the ancestral mitchdondrial type; orange line—the frequency of uninfected individuals with the ancestral mitochondrial type; green line—the frequency of infected individuals with the new mitoype; red line—the frequency of uninfected individuals with the new mitoype. The dashed line represents the baseline equilibrium frequency of the symbiont in the presence of resident mitotype alone. (Online version in colour.)

The association of mitotype and symbionts is also expected to impact symbiont diversity. Where positively selected mitotypes arise in a symbiont-infected individual, our model indicates that these episodes of selection result in the fixation of the symbiont strain originally associated with the beneficial mitotype. Thus, just as the level of mitochondrial diversity may be a function of the history of selection on the symbiont, so symbiont diversity may reflect historical selection on mtDNA. Notably, even weakly selected mitochondria are expected to impact symbiont diversity where these arise initially in an individual carrying a symbiont. Thus, interpretation of a population in which there is a lack of symbiont/mtDNA diversity requires more nuance than the traditional view in which symbiont evolutionary processes dominate the evolution of a neutral marker. A lack of variation in mtDNA and symbionts may reflect a selective sweep event on the symbiont, or a sweep event on the mtDNA.

Finally, our results indicate a further way in which symbionts may affect mitochondrial evolution. Weakly advantageous novel mitotypes will only spread where the mutation occurs in a symbiont-infected lineage. Where symbionts are at prevalence values less than 20%, these mutations are likely to occur in uninfected individuals and then fail to spread, or in the case of a symbiont with the frequency-dependent benefit, spread but be restricted to the uninfected fraction of the population. Thus, the presence of heritable symbionts may increase the time before beneficial mitotypes spread, as weakly beneficial mutations can only invade if they occur initially in symbiont-infected individuals. This process is conceptually similar to background selection for linked nuclear loci, where weakly beneficial mutations do not spread if they initially occur near deleterious mutations [34].

The conclusions of our models depend on the assumption that symbiont and mitochondria are co-transmitted through maternal inheritance. This conclusion is validated in part by

the many observations of sweeps of mtDNA associated with *Wolbachia* spread. Nevertheless, co-transmission may be disrupted through rare paternal mtDNA transmission [35], or intraspecific infectious transmission of the symbiont. Phylogenomic analysis comparing mtDNA and symbiont phylogenies indicates a high level of concordance within species, although some discordance is commonly observed [36–38]. In general, the low level observed is unlikely to disrupt the processes we have been modelling, as the processes we are examining are occurring on short population genetic timescales (compared to the long timescales captured in phylogenomic analyses). Nevertheless, there are two records where symbiont infectious transmission is detectable in the timescale of laboratory experiments [39,40], and some species examined show no concordance of mtDNA and symbiont phylogeny [38]. In these cases, our model assumptions are likely to be sufficiently compromised that the process we have predicted would not be manifested.

The model presented examines the impact of selection for a novel mitotype in a population where hosts either carry a particular symbiont or are uninfected. In natural populations, it is common to find multiple circulating symbiont strains within a species, either as single infections (one symbiont strain/host, e.g. [41]) or as co-infections. For instance, in many cases, a facultative symbiont will co-reside with an obligate one (e.g. [42]). It is expected that selection for beneficial mitotypes will alter symbiont population biology in these cases, with the initially infected type gaining an advantage over other types. Where there are obligate symbionts in addition to a facultative one, the outcome is expected to be similar to that modelled, with the additional observation that obligate symbiont genetic diversity may be reduced, as the strain initially associated with the beneficial mitochondrion is expected to be favoured. More complex model formulations will be required to examine the dynamics where there are multiple strains circulating as single infections.

In conclusion, the coinheritance of mtDNA and maternally inherited symbionts means that selection on mtDNA can alter the dynamics and diversity of the symbiont, both in terms of reducing symbiont diversity within the population and driving the loss of low prevalence facultative symbionts. In female heterogametic taxa, these processes would extend to the associated W chromosome [43] and therefore be even more widespread. A selective sweep in the W chromosome originating in a symbiont-uninfected individual would directly parallel the dynamics of a novel beneficial mitotype. This logic leads to the prediction that symbionts in female heterogametic hosts (e.g. Lepidoptera) would be more commonly subject to these turnover events than comparators in male-heterogametic taxa. The thesis from this study is that the evolution of cytoplasmically inherited components of the genome requires consideration of selection occurring in any and all components.

**Ethics.** This work is *in silico*, with no ethics requirements.

**Data accessibility.** No data were generated in this project. Code for simulations can be found at: https://doi.org/10.6084/m9.figshare.c. 5354999.v2. Supplementary information are provided in the electronic supplementary material [44].

**Authors' contributions.** A.F.: formal analysis, investigation, methodology, writing-original draft, writing-review and editing; M.F.C.: conceptualization, investigation, writing-original draft, writing-review and

editing; G.D.H.: conceptualization, funding acquisition, investigation, methodology, project administration, writing-original draft, writing-review and editing.

All authors gave final approval for publication and agreed to be held accountable for the work performed therein.

Competing interests. We declare no competing interests.

Funding. This work was supported by the UKRI (grant no. NE/S012346/1).

Acknowledgements. We thank three anonymous reviewers for insightful and helpful comments on the manuscript.

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
