## [Peer Review File · Proceedings of the Royal Society B: Biological Sciences]

Review History

RSPB-2021-0722.R0 (Original submission)

Review form: Reviewer 1 (Amanda Brown)

Recommendation

Accept with minor revision (please list in comments)

Scientific importance: Is the manuscript an original and important contribution to its field?

Excellent

General interest: Is the paper of sufficient general interest?

Good

Quality of the paper: Is the overall quality of the paper suitable?

Excellent

Is the length of the paper justified?

Yes

Should the paper be seen by a specialist statistical reviewer?

No

Do you have any concerns about statistical analyses in this paper? If so, please specify them explicitly in your report.

No

It is a condition of publication that authors make their supporting data, code and materials available - either as supplementary material or hosted in an external repository. Please rate, if applicable, the supporting data on the following criteria.

Is it accessible?

Yes

Is it clear?

Yes

Is it adequate?

Yes

Do you have any ethical concerns with this paper?

No

Comments to the Author

Fenton et al examine a novel and truly interesting question, of broad importance, on the topic of cytoplasmic symbionts examining mitochondrial selection as it could impact facultative heritable symbionts, which are well-studied and of widespread importance globally, including examples like *Wolbachia* that affect disease vectors like mosquitoes.

Their manuscript describes results involving three mathematical models that are fairly simple, and clearly explained, without being overly complex or opaque to the biological reader. The models are described clearly in the Appendix. Their conclusion is formulated to suggest these results point to a need to reassess the assumption that the direction of the effects of interacting mitochondria and symbionts flows one way. Instead, their results present model outcomes indicating a perspective shift from regarding the symbiont as driver of the evolution of mtDNA (and mtDNA markers) to instead see that mitochondrial selection drives changes (e.g., elimination or genotype sweeps) in the symbionts.

The paper is succinct and presents novel, quality model results, however, some of the text and figures are unclear. Most importantly, the goals and biological scenarios of the models – and the premise – is missing or under-explained throughout the text and figures, except for the final half of the last paragraph of the introduction. The later sections could all be improved by re-iterating some of the text between lines 71 and 82 within the Results and Discussion sections, and even within the Figures and Abstract, to really make more plain the purpose of the models.

Major comments, line-by-line:

Abstract: the abstract is too vague about methods; authors should modify, perhaps in line 24, one or two key components of the model(s) it/themselves.

Lines 83 – 107: The modelling framework section is not sufficiently detailed. While journal style might limit the length and amount of detail allowed or preferred here, it is inadequate to relegate not just all mathematical details to the Appendix, but all the underlying explorations and their goals. At a minimum, the models (i) through (iii) should be stated here, for completeness and clarity for the reader. Furthermore, given that this math is required to understand the model results and figures, I suggest judiciously moving all or part of (i) recursion equations and/or the (i) system single equation and/or the (i) four equilibria equations into the main framework section; and similarly or in parallel, doing the same for (ii) and (iii).

Lines 116-120: Something is missing in this sentence; what is meant by “all others” – other what?

Line 129: What do the authors specifically define as “low prevalence” here? The figure and model can be seen, but the number range of “low” here should be elaborated somewhat more in the text, for clarity.

Lines 132-145: This model is very well-described and is a highlight of the manuscript.

Lines 153-154: It is almost clear how/why the mitotype variation becomes partitioned, but it would help here to either explain this more, or provide a figure or some more full conceptual framing, as this point seems very interesting and a key point of this model

Line 146: Compared to the previous paragraph, this paragraph is a little bit difficult to follow. To start with, the “special case of no segregational loss of the symbiont” is biologically opaque. Can this be stated first in more plain (lay biologist) language in the context of organisms and their life history/ecology?

Line 189-190: it's not really clear why the final scenario in this model is not shown. Presumably there is something trivial about this result? It might help to explain a little bit further here.

Discussion:

Line 226: need a reference here to support that studies show mtDNA is not a neutral marker

Lines 226-228: Although the grammar is somewhat in need of correction in this sentence (see comment below), this statement that the process modelled here occurs commonly is unclear. If by “commonly”, the authors mean to refer to the model outcomes under different starting states, this should be reiterated here more plainly and fully for the reader, as this outcome has, by now, become a little bit lost or obscure. I suggest stating the cases that make this “common”.

Lines 229-230: this seems to need a citation from the literature.

Lines 239-240: this sentence reads as though it arises from empirical observations, but surely it's meant to be the model outcome, right? If so, this sentence, and others like it should more explicitly state (at least in the Discussion – though perhaps not always in the Results) that this is the model's prediction. E.g., “Our model suggests that where positively selected mitotypes arise in a symbiont infected....”

Lines 265-270: A comment, not necessarily a criticism of the models, here is that much of the framing of this manuscript's approach would not apply in most cases to parasitoid wasp symbiont systems, in which there seem to be other scenarios for symbiont horizontal exchange.

Lines 273-276: While the discussion of W chromosomes is warranted and of interest, it is treated too briefly here, and could use at least one more sentence to clarify or explore fully the idea. Also, the final sentence seems disjointed from the second-last sentence, to the point that I'm not fully sure that I understand the authors here: which cytoplasmically inherited component of which gene? Do they mean a gene on the W chromosome? What do the authors mean by “any and all components”? Components of what?

Figure 1: t is not adequately defined in the legend; authors should re-state in the figure legends the material of the text that is necessary to understand the figures – i.e. positively selected mitochondria $t > 0$. B is not adequately defined in the legend. Mu is not defined.

Figure 2 Legend: B is not defined adequately in this legend. It is the y-intersect, and all else seems to be the same – but you need to explain in the legend that it is the net benefit of symbiont infection in model (i). The heatmap scale gradient looks the same in all cases – is it supposed to be

the same? What value does the heatmap add to this?

Figure 3: use of hyphen between symbiont and uninfected is inconsistent between this figure and figure 2 – choose which form to use and use it consistently. Again, just as B was not defined

Figure 5: Good figure, but again, the legend has some terms/acronyms that need a little more explanation. CI is mentioned at the end but should be defined formally in the legend as cytoplasmic incompatibility.

Minor line by line comments:

Line 43: “both as” ... ??? do the authors mean that these microbes always serve both roles? I think this should be changed to “either as”.

Line 53: perhaps add commas around “in part”, i.e., is a product, in part, of the ...

Lines 97-100: font within curly braces differs from font for variables defined in preceding sentences

Line 152: suggest changing “symbiont uninfected” to “symbiont-uninfected”, and similarly throughout

Line 171: Is “dripping” the best word here?

Line 209: change “beside” to “besides”

Lines 214-215: something is wrong with this sentence grammatically – should there be a noun after “if” and another noun after “carries” – should “leave” be “and leaves”?

Line 216: to what does “this” in “this only occurs” refer? I can’t follow.

Lines 226-228: again, something is wrong with this sentence grammatically at “these data support the process modelled here occurring sufficiently”; which process modelled here?

Line 255: change “conceptual” to “conceptually”

Lines 275-276: this last sentence of the paper has some grammar issue. Perhaps add “the” before “cytoplasmically” or add “s” to “component”.

Review form: Reviewer 2

Recommendation

Accept with minor revision (please list in comments)

Scientific importance: Is the manuscript an original and important contribution to its field?

Excellent

General interest: Is the paper of sufficient general interest?

Excellent

Quality of the paper: Is the overall quality of the paper suitable?

Excellent

Is the length of the paper justified?

Yes

Should the paper be seen by a specialist statistical reviewer?

Yes

Do you have any concerns about statistical analyses in this paper? If so, please specify them explicitly in your report.

No

It is a condition of publication that authors make their supporting data, code and materials available - either as supplementary material or hosted in an external repository. Please rate, if applicable, the supporting data on the following criteria.

Is it accessible?

Yes

Is it clear?

Yes

Is it adequate?

Yes

Do you have any ethical concerns with this paper?

No

Comments to the Author

The authors present a very interesting study on coevolutionary dynamics between mitochondria and facultative heritable symbionts in arthropods and show that under certain conditions the introduction of a new beneficial mitotype can lead to loss of symbiont variation from the population. These results have important implications for our general understanding of heritable symbioses. I found the paper well written and the analyses appropriately conducted. There were a few sections in the manuscript that could benefit from further clarification and inclusion of additional discussion points. Please see my comments below.

- Since this paper only addresses facultative heritable symbioses, this needs to be pointed out more strongly in the introduction and clearly delineated from obligate heritable symbioses, which are very common in insects. Especially for the unfamiliar reader I suggest to include another paragraph that explains the differences between these symbiotic types. Also, do you refer here to facultative symbionts as primary symbionts or do you consider the scenario where they are secondary symbionts to obligate symbionts? I think all of these aspects need to be explained a bit more to avoid confusion about the implications of your results.
- In this context, how would the presence of an obligate symbionts influence the evolutionary dynamics that you are describing? Can you discuss this a little bit as it is relevant to many insect symbioses?
- Can the model be applied to symbiotic systems other than arthropods? It would be great if you could address this aspect or otherwise compare with different symbiotic systems
- L70: Needs to clearly state that model refers to interactions between mitochondria and bacterial symbionts
- L72: What do you mean with direct drive phenotype? I see you describe this later in the discussion, I would already explain it here
- L92-94: This sentence is slightly confusing. Can you please rewrite to clearly state the limitations of your model and which aspects cannot be accounted for?
- L102 and 468: Which values of t were tested to reflect different strengths of selection? In the figures it looks like t goes up to 0.2 not 0.02?
- L108ff: I found it rather hard to remember the main findings from the results. I would

suggest rewriting this a little bit and adding a summarizing sentence at the end of each section

- L220-221: Write out full species names
- L229: Lacks references. Can you provide some examples?
- Figures need higher resolution

Review form: Reviewer 3

Recommendation

Accept with minor revision (please list in comments)

Scientific importance: Is the manuscript an original and important contribution to its field?

Excellent

General interest: Is the paper of sufficient general interest?

Excellent

Quality of the paper: Is the overall quality of the paper suitable?

Excellent

Is the length of the paper justified?

Yes

Should the paper be seen by a specialist statistical reviewer?

No

Do you have any concerns about statistical analyses in this paper? If so, please specify them explicitly in your report.

No

It is a condition of publication that authors make their supporting data, code and materials available - either as supplementary material or hosted in an external repository. Please rate, if applicable, the supporting data on the following criteria.

Is it accessible?

Yes

Is it clear?

Yes

Is it adequate?

Yes

Do you have any ethical concerns with this paper?

No

Comments to the Author

The authors investigate the effect of a novel, beneficial mitotype on the distribution of maternally-inherited symbionts. They do this for three different possible symbiont types: symbionts that have a constant beneficial effect on their host, symbionts with a negative frequency-dependent benefit, and symbionts with a positive frequency-dependent benefit via cytoplasmic incompatibility. The authors find that when the mitotype arises in an individual infected with the symbiont, the new mitotype nearly always goes to fixation. However, when the new mitotype arises in an uninfected individual, a wider variety of outcomes are possible. The

new mitotype can drive the symbiont out of the population, or the new mitotype itself can be lost. Occasionally, there is also an equilibrium between hosts with the old mitotype (some or all of which are infected) and uninfected hosts with the new mitotype. The exact regions of parameter space that cause each outcome to occur depend strongly on the type of benefit the symbiont confers on its host.

This paper investigates a likely common influence on symbiont population dynamics that hasn't been modeled before. The authors' investigation of the effects on multiple types of symbionts is likely to make this interesting to a wide variety of readers: both those interested in mitochondrial effects and those interested in understanding how symbiont and host traits interact to determine symbiont population dynamics. The finding that rare symbionts are more likely to be lost due to novel beneficial mitotypes is intriguing, as are the cases where the presence of the symbiont can allow the resident and novel mitotypes to coexist. The paper is also very well-written, and the results section in particular is very clear.

I just have some small comments:

For Figure 5, I am very curious about the regions that look like they are not completely blue or yellow. Is there coexistence when the symbiont cannot persist and t is slightly greater than 0? Or when $h = 0.1$ and $t \sim 0.065$, $\mu \sim 0.02$? I would love to read a bit of intuition about what is going on in those regions. I'm also curious why the novel mitotype doesn't seem to fully invade (or invade at all?) when t is small but nonzero and the symbiont is absent from the population.

The appendix is very well-written, and the Mathematica code is also very clear. The tables giving offspring produced by each parental type were particularly nice for understanding the equations. The one point where I had trouble was in understanding why certain equilibria were biologically relevant or not. Some were obvious, but there were a few of the more complicated expressions that I wasn't able to figure out. It might be helpful to add some code to the Mathematica files that sorts the equilibria into the biologically relevant and irrelevant ones.

In Figures 2 and 3, I think it might be helpful to make some of the lines dashed or dotted to distinguish them more easily. This would be especially nice for the $b = 0.02$ case in Figure 3.

In Figure 2: When I initially looked at the region where the symbiont can persist and the novel mitotype can invade, I thought those two things could happen simultaneously. The rest of the paper makes it clear that the novel mitotype drives the symbiont extinct, but it might still be worth adding a note about this to the legend.

Figure 3 legend: I think there is a typo. It says $c = 0.05$, but in that case the symbiont would not be able to persist in the population when $b = 0.02$ or 0.04 .

In Figures 1 and 4, it would be nice to have a label for the dashed line.

I think there are a couple typos in the equations:

Line 571: I don't get this expression when I set the equation on line 570 to 1 and solve for t .

Line 603: I think there might be an extra h in the denominator. Should it be $1 - h pT (1 - pT (1 - \mu))$?

Lines 606 and 607: I think the first term in the numerator (currently h) should be \sqrt{h} .

Line 189: Should $t > 1$ should be $t > 0$?

Decision letter (RSPB-2021-0722.R0)

10-Jun-2021

Dear Dr Hurst:

I am writing to inform you that your manuscript RSPB-2021-0722 entitled "Positive selection on mitochondria may eliminate heritable microbes from arthropod populations." has, in its current form, been rejected for publication in Proceedings B.

This action has been taken on the advice of referees and the Associate Editor, who all found the study interesting but have also recommended that revisions are necessary before we can proceed. With this in mind we would be happy to consider a resubmission, provided the comments of the referees are fully addressed. However please note that this is not a provisional acceptance.

Sincerely,
Professor Hans Heesterbeek
<mailto:proceedingsb@royalsociety.org>

Associate Editor
Comments to Author:

This is an interesting theoretical study providing new insights into the field. All reviewers and myself found this a very interesting paper. The reviewers made some important and useful comments that may help improve the manuscript further. Most importantly, I concur with Reviewer 1 that in the model description section, more mathematical detail about the three different scenarios is needed. While it may be too space-intensive to present all recursions, the key equilibria should be shown in the main text. The main text needs to be understandable as an independent unit and this is currently not the case. Similarly, Reviewer 2 asked for some more details about how the limitation to facultative heritable symbioses affects the conclusions and how these differ from the obligate heritable symbioses.

Reviewer(s)' Comments to Author:

Referee: 1

Comments to the Author(s)

Fenton et al examine a novel and truly interesting question, of broad importance, on the topic of cytoplasmic symbionts examining mitochondrial selection as it could impact facultative heritable symbionts, which are well-studied and of widespread importance globally, including examples like *Wolbachia* that affect disease vectors like mosquitoes.

Their manuscript describes results involving three mathematical models that are fairly simple, and clearly explained, without being overly complex or opaque to the biological reader. The models are described clearly in the Appendix. Their conclusion is formulated to suggest these results point to a need to reassess the assumption that the direction of the effects of interacting mitochondria and symbionts flows one way. Instead, their results present model outcomes indicating a perspective shift from regarding the symbiont as driver of the evolution of mtDNA (and mtDNA markers) to instead see that mitochondrial selection drives changes (e.g., elimination or genotype sweeps) in the symbionts.

The paper is succinct and presents novel, quality model results, however, some of the text and figures are unclear. Most importantly, the goals and biological scenarios of the models – and the premise – is missing or under-explained throughout the text and figures, except for the final half of the last paragraph of the introduction. The later sections could all be improved by re-iterating some of the text between lines 71 and 82 within the Results and Discussion sections, and even within the Figures and Abstract, to really make more plain the purpose of the models.

Major comments, line-by-line:

Abstract: the abstract is too vague about methods; authors should modify, perhaps in line 24, one or two key components of the model(s) it/themselves.

Lines 83 – 107: The modelling framework section is not sufficiently detailed. While journal style might limit the length and amount of detail allowed or preferred here, it is inadequate to relegate not just all mathematical details to the Appendix, but all the underlying explorations and their goals. At a minimum, the models (i) through (iii) should be stated here, for completeness and clarity for the reader. Furthermore, given that this math is required to understand the model results and figures, I suggest judiciously moving all or part of (i) recursion equations and/or the (i) system single equation and/or the (i) four equilibria equations into the main framework section; and similarly or in parallel, doing the same for (ii) and (iii).

Lines 116-120: Something is missing in this sentence; what is meant by “all others” – other what?

Line 129: What do the authors specifically define as “low prevalence” here? The figure and model can be seen, but the number range of “low” here should be elaborated somewhat more in the text, for clarity.

Lines 132-145: This model ii is very well-described and is a highlight of the manuscript.

Lines 153-154: It is almost clear how/why the mitotype variation becomes partitioned, but it would help here to either explain this more, or provide a figure or some more full conceptual framing, as this point seems very interesting and a key point of this model

Line 146: Compared to the previous paragraph, this paragraph is a little bit difficult to follow. To start with, the “special case of no segregational loss of the symbiont” is biologically opaque. Can this be stated first in more plain (lay biologist) language in the context of organisms and their life history/ecology?

Line 189-190: it's not really clear why the final scenario in this model is not shown. Presumably there is something trivial about this result? It might help to explain a little bit further here.

Discussion:

Line 226: need a reference here to support that studies show mtDNA is not a neutral marker

Lines 226-228: Although the grammar is somewhat in need of correction in this sentence (see comment below), this statement that the process modelled here occurs commonly is unclear. If by "commonly", the authors mean to refer to the model outcomes under different starting states, this should be reiterated here more plainly and fully for the reader, as this outcome has, by now, become a little bit lost or obscure. I suggest stating the cases that make this "common".

Lines 229-230: this seems to need a citation from the literature.

Lines 239-240: this sentence reads as though it arises from empirical observations, but surely it's meant to be the model outcome, right? If so, this sentence, and others like it should more explicitly state (at least in the Discussion – though perhaps not always in the Results) that this is the model's prediction. E.g., "Our model suggests that where positively selected mitotypes arise in a symbiont infected...."

Lines 265-270: A comment, not necessarily a criticism of the models, here is that much of the framing of this manuscript's approach would not apply in most cases to parasitoid wasp symbiont systems, in which there seem to be other scenarios for symbiont horizontal exchange.

Lines 273-276: While the discussion of W chromosomes is warranted and of interest, it is treated too briefly here, and could use at least one more sentence to clarify or explore fully the idea. Also, the final sentence seems disjointed from the second-last sentence, to the point that I'm not fully sure that I understand the authors here: which cytoplasmically inherited component of which gene? Do they mean a gene on the W chromosome? What do the authors mean by "any and all components"? Components of what?

Figure 1: t is not adequately defined in the legend; authors should re-state in the figure legends the material of the text that is necessary to understand the figures – i.e. positively selected mitochondria $t > 0$. B is not adequately defined in the legend. Mu is not defined.

Figure 2 Legend: B is not defined adequately in this legend. It is the y-intersect, and all else seems to be the same – but you need to explain in the legend that it is the net benefit of symbiont infection in model (i). The heatmap scale gradient looks the same in all cases – is it supposed to be the same? What value does the heatmap add to this?

Figure 3: use of hyphen between symbiont and uninfected is inconsistent between this figure and figure 2 – choose which form to use and use it consistently. Again, just as B was not defined

Figure 5: Good figure, but again, the legend has some terms/acronyms that need a little more explanation. CI is mentioned at the end but should be defined formally in the legend as cytoplasmic incompatibility.

Minor line by line comments:

Line 43: "both as" ... ??? do the authors mean that these microbes always serve both roles? I think this should be changed to "either as".

Line 53: perhaps add commas around "in part", i.e., is a product, in part, of the ...

Lines 97-100: font within curly braces differs from font for variables defined in preceding sentences

Line 152: suggest changing “symbiont uninfected” to “symbiont-uninfected”, and similarly throughout

Line 171: Is “dripping” the best word here?

Line 209: change “beside” to “besides”

Lines 214-215: something is wrong with this sentence grammatically – should there be a noun after “if” and another noun after “carries” – should “leave” be “and leaves”?

Line 216: to what does “this” in “this only occurs” refer? I can’t follow.

Lines 226-228: again, something is wrong with this sentence grammatically at “these data support the process modelled here occurring sufficiently”; which process modelled here?

Line 255: change “conceptual” to “conceptually”

Lines 275-276: this last sentence of the paper has some grammar issue. Perhaps add “the” before “cytoplasmically” or add “s” to “component”.

Referee: 2

Comments to the Author(s)

The authors present a very interesting study on coevolutionary dynamics between mitochondria and facultative heritable symbionts in arthropods and show that under certain conditions the introduction of a new beneficial mitotype can lead to loss of symbiont variation from the population. These results have important implications for our general understanding of heritable symbioses. I found the paper well written and the analyses appropriately conducted. There were a few sections in the manuscript that could benefit from further clarification and inclusion of additional discussion points. Please see my comments below.

- Since this paper only addresses facultative heritable symbioses, this needs to be pointed out more strongly in the introduction and clearly delineated from obligate heritable symbioses, which are very common in insects. Especially for the unfamiliar reader I suggest to include another paragraph that explains the differences between these symbiotic types. Also, do you refer here to facultative symbionts as primary symbionts or do you consider the scenario where they are secondary symbionts to obligate symbionts? I think all of these aspects need to be explained a bit more to avoid confusion about the implications of your results.
- In this context, how would the presence of an obligate symbionts influence the evolutionary dynamics that you are describing? Can you discuss this a little bit as it is relevant to many insect symbioses?
- Can the model be applied to symbiotic systems other than arthropods? It would be great if you could address this aspect or otherwise compare with different symbiotic systems
- L70: Needs to clearly state that model refers to interactions between mitochondria and bacterial symbionts
- L72: What do you mean with direct drive phenotype? I see you describe this later in the discussion, I would already explain it here
- L92-94: This sentence is slightly confusing. Can you please rewrite to clearly state the limitations of your model and which aspects cannot be accounted for?
- L102 and 468: Which values of t were tested to reflect different strengths of selection? In the figures it looks like t goes up to 0.2 not 0.02?
- L108ff: I found it rather hard to remember the main findings from the results. I would suggest rewriting this a little bit and adding a summarizing sentence at the end of each section

- L220-221: Write out full species names
- L229: Lacks references. Can you provide some examples?
- Figures need higher resolution

Referee: 3

Comments to the Author(s)

The authors investigate the effect of a novel, beneficial mitotype on the distribution of maternally-inherited symbionts. They do this for three different possible symbiont types: symbionts that have a constant beneficial effect on their host, symbionts with a negative frequency-dependent benefit, and symbionts with a positive frequency-dependent benefit via cytoplasmic incompatibility. The authors find that when the mitotype arises in an individual infected with the symbiont, the new mitotype nearly always goes to fixation. However, when the new mitotype arises in an uninfected individual, a wider variety of outcomes are possible. The new mitotype can drive the symbiont out of the population, or the new mitotype itself can be lost. Occasionally, there is also an equilibrium between hosts with the old mitotype (some or all of which are infected) and uninfected hosts with the new mitotype. The exact regions of parameter space that cause each outcome to occur depend strongly on the type of benefit the symbiont confers on its host.

This paper investigates a likely common influence on symbiont population dynamics that hasn't been modeled before. The authors' investigation of the effects on multiple types of symbionts is likely to make this interesting to a wide variety of readers: both those interested in mitochondrial effects and those interested in understanding how symbiont and host traits interact to determine symbiont population dynamics. The finding that rare symbionts are more likely to be lost due to novel beneficial mitotypes is intriguing, as are the cases where the presence of the symbiont can allow the resident and novel mitotypes to coexist. The paper is also very well-written, and the results section in particular is very clear.

I just have some small comments:

For Figure 5, I am very curious about the regions that look like they are not completely blue or yellow. Is there coexistence when the symbiont cannot persist and t is slightly greater than 0? Or when $h = 0.1$ and $t \sim 0.065$, $\mu \sim 0.02$? I would love to read a bit of intuition about what is going on in those regions. I'm also curious why the novel mitotype doesn't seem to fully invade (or invade at all?) when t is small but nonzero and the symbiont is absent from the population.

The appendix is very well-written, and the Mathematica code is also very clear. The tables giving offspring produced by each parental type were particularly nice for understanding the equations. The one point where I had trouble was in understanding why certain equilibria were biologically relevant or not. Some were obvious, but there were a few of the more complicated expressions that I wasn't able to figure out. It might be helpful to add some code to the Mathematica files that sorts the equilibria into the biologically relevant and irrelevant ones.

In Figures 2 and 3, I think it might be helpful to make some of the lines dashed or dotted to distinguish them more easily. This would be especially nice for the $b = 0.02$ case in Figure 3.

In Figure 2: When I initially looked at the region where the symbiont can persist and the novel mitotype can invade, I thought those two things could happen simultaneously. The rest of the paper makes it clear that the novel mitotype drives the symbiont extinct, but it might still be worth adding a note about this to the legend.

Figure 3 legend: I think there is a typo. It says $c = 0.05$, but in that case the symbiont would not be able to persist in the population when $b = 0.02$ or 0.04 .

In Figures 1 and 4, it would be nice to have a label for the dashed line.

I think there are a couple typos in the equations:

Line 571: I don't get this expression when I set the equation on line 570 to 1 and solve for t.

Line 603: I think there might be an extra h in the denominator. Should it be $1 - h pT (1 - pT (1 - \mu))$?

Lines 606 and 607: I think the first term in the numerator (currently h) should be \sqrt{h} .

Line 189: Should $t > 1$ should be $t > 0$?

Author's Response to Decision Letter for (RSPB-2021-0722.R0)

See Appendix A.

RSPB-2021-1735.R0

Review form: Reviewer 1 (Amanda Brown)

Recommendation

Accept as is

Scientific importance: Is the manuscript an original and important contribution to its field?

Excellent

General interest: Is the paper of sufficient general interest?

Excellent

Quality of the paper: Is the overall quality of the paper suitable?

Excellent

Is the length of the paper justified?

Yes

Should the paper be seen by a specialist statistical reviewer?

No

Do you have any concerns about statistical analyses in this paper? If so, please specify them explicitly in your report.

No

It is a condition of publication that authors make their supporting data, code and materials available - either as supplementary material or hosted in an external repository. Please rate, if applicable, the supporting data on the following criteria.

Is it accessible?

Yes

Is it clear?

Yes

Is it adequate?

Yes

Do you have any ethical concerns with this paper?

No

Comments to the Author

This is my second review of this manuscript, after the authors' revisions. I have looked in detail at the authors' revision as a whole, and in line-by-line changes in response to my previous review and those of the other two reviewers, and I feel the manuscript is now much improved and makes a significant (and now much clearer) contribution to the field of evolutionary models of heritable microbes. The research is novel and exciting and the approach is presented thoroughly and clearly.

Review form: Reviewer 2

Recommendation

Accept as is

Scientific importance: Is the manuscript an original and important contribution to its field?

Excellent

General interest: Is the paper of sufficient general interest?

Excellent

Quality of the paper: Is the overall quality of the paper suitable?

Excellent

Is the length of the paper justified?

Yes

Should the paper be seen by a specialist statistical reviewer?

Yes

Do you have any concerns about statistical analyses in this paper? If so, please specify them explicitly in your report.

No

It is a condition of publication that authors make their supporting data, code and materials available - either as supplementary material or hosted in an external repository. Please rate, if applicable, the supporting data on the following criteria.

Is it accessible?

Yes

Is it clear?

Yes

Is it adequate?

Yes

Do you have any ethical concerns with this paper?

No

Comments to the Author

Thank you very much for revising the manuscript based on my comments. I am satisfied with the current version and would like to compliment the authors on a fascinating study.

Review form: Reviewer 3**Recommendation**

Accept with minor revision (please list in comments)

Scientific importance: Is the manuscript an original and important contribution to its field?

Excellent

General interest: Is the paper of sufficient general interest?

Excellent

Quality of the paper: Is the overall quality of the paper suitable?

Excellent

Is the length of the paper justified?

Yes

Should the paper be seen by a specialist statistical reviewer?

No

Do you have any concerns about statistical analyses in this paper? If so, please specify them explicitly in your report.

No

It is a condition of publication that authors make their supporting data, code and materials available - either as supplementary material or hosted in an external repository. Please rate, if applicable, the supporting data on the following criteria.

Is it accessible?

Yes

Is it clear?

Yes

Is it adequate?

Yes

Do you have any ethical concerns with this paper?

No

Comments to the Author

The authors are right about the math that was on lines 606 and 607 (now lines 176 and 177). I think the expression on line 680 may still have a typo -- it looks to me like the whole thing should be divided by 2. Otherwise, they addressed all my comments, and I think it is a very exciting model.

Decision letter (RSPB-2021-1735.R0)

20-Aug-2021

Dear Dr Hurst

I am pleased to inform you that your very nice manuscript RSPB-2021-1735 entitled "Positive selection on mitochondria may eliminate heritable microbes from arthropod populations." has been accepted for publication in Proceedings B.

The referees have all recommended publication, but one reviewer suggests some minor revision (a potential typo) to your manuscript. Therefore, I invite you to respond to the referee's comment and revise your manuscript. Because the schedule for publication is very tight, it is a condition of publication that you submit the revised version of your manuscript within 7 days. If you do not think you will be able to meet this date please let us know.

Sincerely,
Professor Hans Heesterbeek
mailto: proceedingsb@royalsociety.org

Associate Editor
Comments to Author:

The authors have made a great effort to improve the overall structure and clarity of their manuscript and now present the right amount of mathematical detail to allow readers to understand where the conclusions are coming from.

Reviewer(s)' Comments to Author:

Referee: 2

Comments to the Author(s).

Thank you very much for revising the manuscript based on my comments. I am satisfied with the current version and would like to compliment the authors on a fascinating study.

Referee: 1

Comments to the Author(s).

This is my second review of this manuscript, after the authors' revisions. I have looked in detail at the authors' revision as a whole, and in line-by-line changes in response to my previous review and those of the other two reviewers, and I feel the manuscript is now much improved and makes a significant (and now much clearer) contribution to the field of evolutionary models of

heritable microbes. The research is novel and exciting and the approach is presented thoroughly and clearly.

Referee: 3

Comments to the Author(s).

The authors are right about the math that was on lines 606 and 607 (now lines 176 and 177). I think the expression on line 680 may still have a typo -- it looks to me like the whole thing should be divided by 2. Otherwise, they addressed all my comments, and I think it is a very exciting model.

Author's Response to Decision Letter for (RSPB-2021-1735.R0)

See Appendix B.

Decision letter (RSPB-2021-1735.R1)

02-Sep-2021

Dear Dr Hurst

I am pleased to inform you that your manuscript entitled "Positive selection on mitochondria may eliminate heritable microbes from arthropod populations." has been accepted for publication in Proceedings B.

Your article has been estimated as being 9 pages long. Our Production Office will be able to confirm the exact length at proof stage.

Data Accessibility section

Open Access

Paper charges

Sincerely,

Proceedings B

Appendix A

We thank the reviewers for their comments, which have proven helpful in improving the manuscript. We detail our actions in response to the reviewer comments below – the original reviewer comments are in plain text, and our reply in italics.

Reviewer(s)' Comments to Author:

Referee: 1

The paper is succinct and presents novel, quality model results, however, some of the text and figures are unclear. Most importantly, the goals and biological scenarios of the models – and the premise – is missing or under-explained throughout the text and figures, except for the final half of the last paragraph of the introduction. The later sections could all be improved by re-iterating some of the text between lines 71 and 82 within the Results and Discussion sections, and even within the Figures and Abstract, to really make more plain the purpose of the models.

Major comments, line-by-line:

Abstract: the abstract is too vague about methods; authors should modify, perhaps in line 24, one or two key components of the model(s) it/themselves.

This is a helpful suggestion – we have done this to improve the abstract. However, we are limited by a 200 word limit.

Lines 83 – 107: The modelling framework section is not sufficiently detailed. While journal style might limit the length and amount of detail allowed or preferred here, it is inadequate to relegate not just all mathematical details to the Appendix, but all the underlying explorations and their goals. At a minimum, the models (i) through (iii) should be stated here, for completeness and clarity for the reader. Furthermore, given that this math is required to understand the model results and figures, I suggest judiciously moving all or part of (i) recursion equations and/or the (i) system single equation and/or the (i) four equilibria equations into the main framework section; and similarly or in parallel, doing the same for (ii) and (iii).

We have now expanded the Methods section to describe each model in greater depth, moving all the recursion equations there from the Appendix, along with the analyses of the conditions (equations, equilibria and stability criteria) for the symbiont to persist in the presence of the resident mitotype alone. We choose not to include the equilibria of the full system here, leaving them in the Appendix as previously, because we feel these take up a lot of extra space, and the mathematical details aren't essential to understand the corresponding figures.

Lines 116-120: Something is missing in this sentence; what is meant by “all others” – other what?
We have now specified the alternate cytoplasmic combinations we are comparing to.

Line 129: What do the authors specifically define as “low prevalence” here? The figure and model can be seen, but the number range of “low” here should be elaborated somewhat more in the text, for clarity.

We have placed a quantity against this now for clarity.

Lines 132-145: This model ii is very well-described and is a highlight of the manuscript.

Thanks!

Lines 153-154: It is almost clear how/why the mitotype variation becomes partitioned, but it would help here to either explain this more, or provide a figure or some more full conceptual framing, as this point seems very interesting and a key point of this model.

We have added a little clarification

Line 146: Compared to the previous paragraph, this paragraph is a little bit difficult to follow. To start with, the “special case of no segregational loss of the symbiont” is biologically opaque. Can this be stated first in more plain (lay biologist) language in the context of organisms and their life history/ecology?

Yes, this is best framed in terms of perfect vertical transmission rather than no-segregational loss, so we have done this.

Line 189-190: it's not really clear why the final scenario in this model is not shown. Presumably there is something trivial about this result? It might help to explain a little bit further here.

We have now added a new figure (Fig 6) to illustrate this result.

Discussion:

Line 226: need a reference here to support that studies show mtDNA is not a neutral marker
Added

Lines 226-228: Although the grammar is somewhat in need of correction in this sentence (see comment below), this statement that the process modelled here occurs commonly is unclear. If by “commonly”, the authors mean to refer to the model outcomes under different starting states, this should be reiterated here more plainly and fully for the reader, as this outcome has, by now, become a little bit lost or obscure. I suggest stating the cases that make this “common”.

Both clarified

Lines 229-230: this seems to need a citation from the literature.

Added

Lines 239-240: this sentence reads as though it arises from empirical observations, but surely it's meant to be the model outcome, right? If so, this sentence, and others like it should more explicitly state (at least in the Discussion – though perhaps not always in the Results) that this is the model's prediction. E.g., “Our model suggests that where positively selected mitotypes arise in a symbiont infected....”

Fair comment, amended.

Lines 265-270: A comment, not necessarily a criticism of the models, here is that much of the framing of this manuscript's approach would not apply in most cases to parasitoid wasp symbiont systems, in which there seem to be other scenarios for symbiont horizontal exchange.

There are two records of unusual systems where this occurs – it is not all parasitic wasp/symbiont associations. We have now noted this in the text so the literature context is clear.

Lines 273-276: While the discussion of W chromosomes is warranted and of interest, it is treated too briefly here, and could use at least one more sentence to clarify or explore fully the idea. Also, the final sentence seems disjointed from the second-last sentence, to the point that I'm not fully sure that I understand the authors here: which cytoplasmically inherited component of which gene? Do they mean a gene on the W chromosome? What do the authors mean by "any and all components"? Components of what?

Thank you, expanded and clarified.

Figure 1: t is not adequately defined in the legend; authors should re-state in the figure legends the material of the text that is necessary to understand the figures – i.e. positively selected mitochondria $t > 0$. B is not adequately defined in the legend. μ is not defined.

We now define these parameters explicitly in the legend, and provide additional information about each one

Figure 2 Legend: B is not defined adequately in this legend. It is the y -intercept, and all else seems to be the same – but you need to explain in the legend that it is the net benefit of symbiont infection in model (i). The heatmap scale gradient looks the same in all cases – is it supposed to be the same? What value does the heatmap add to this?

The legend is now clearer – we feel the heatmap is important to allow the reader a view of the prevalence of the symbiont under these conditions, which is not otherwise noted.

Figure 3: use of hyphen between symbiont and uninfected is inconsistent between this figure and figure 2 – choose which form to use and use it consistently.

Have added a hyphen throughout

Figure 3: Again, just as B was not defined

Now defined

Figure 5: Good figure, but again, the legend has some terms/acronyms that need a little more explanation. CI is mentioned at the end but should be defined formally in the legend as cytoplasmic incompatibility.

Have defined CI early in the legend

Minor line by line comments:

Line 43: "both as" ... ??? do the authors mean that these microbes always serve both roles? I think this should be changed to "either as".

modified

Line 53: perhaps add commas around "in part", i.e., is a product, in part, of the ...

Lines 97-100: font within curly braces differs from font for variables defined in preceding sentences

Have standardised fonts for all variables throughout

Line 152: suggest changing "symbiont uninfected" to "symbiont-uninfected", and similarly throughout

Done throughout

Line 171: Is “dripping” the best word here?

Changed

Line 209: change “beside” to “besides”

Changed

Lines 214-215: something is wrong with this sentence grammatically – should there be a noun after “if” and another noun after “carries” – should “leave” be “and leaves”?

Clarified.

Line 216: to what does “this” in “this only occurs” refer? I can’t follow.

This process.

Lines 226-228: again, something is wrong with this sentence grammatically at “these data support the process modelled here occurring sufficiently”; which process modelled here?

done

Line 255: change “conceptual” to “conceptually”

Done

Lines 275-276: this last sentence of the paper has some grammar issue. Perhaps add “the” before “cytoplasmically” or add “s” to “component”.

Done.

Referee: 2

Comments to the Author(s)

The authors present a very interesting study on coevolutionary dynamics between mitochondria and facultative heritable symbionts in arthropods and show that under certain conditions the introduction of a new beneficial mitotype can lead to loss of symbiont variation from the population. These results have important implications for our general understanding of heritable symbioses. I found the paper well written and the analyses appropriately conducted. There were a few sections in the manuscript that could benefit from further clarification and inclusion of additional discussion points. Please see my comments below.

- Since this paper only addresses facultative heritable symbioses, this needs to be pointed out more strongly in the introduction and clearly delineated from obligate heritable symbioses, which are very common in insects. Especially for the unfamiliar reader I suggest to include another paragraph that explains the differences between these symbiotic types. Also, do you refer here to facultative symbionts as primary symbionts or do you consider the scenario where they are secondary symbionts to obligate symbionts? I think all of these aspects need to be explained a bit more to avoid confusion about the implications of your results.

Obligate symbionts can be considered as the case $b > 0$ and $\mu = 0$ – always beneficial with no segregational loss. We didn't discuss this class in any depth as all of our models reflect advantageous mitochondria altering the balance of advantage between symbiont-infected and symbiont uninfected individuals – which don't exist for obligate heritable microbes. We do make clear now the distinction in 1168-76.

- In this context, how would the presence of an obligate symbionts influence the evolutionary dynamics that you are describing? Can you discuss this a little bit as it is relevant to many insect symbioses?

We now discuss this as an aspect for further work in the discussion 11360-370.

- Can the model be applied to symbiotic systems other than arthropods? It would be great if you could address this aspect or otherwise compare with different symbiotic systems
The model technically works for all maternally inheritable microbes, which are found in plants, microeuks and a wide range of invertebrates. We now make this clear in the introduction and abstract 1135-40

- L70: Needs to clearly state that model refers to interactions between mitochondria and bacterial symbionts

Done

- L72: What do you mean with direct drive phenotype? I see you describe this later in the discussion, I would already explain it here

We describe it more now in the following sentence.

- L92-94: This sentence is slightly confusing. Can you please rewrite to clearly state the limitations of your model and which aspects cannot be accounted for?

Done.

- L102 and 468: Which values of t were tested to reflect different strengths of selection? In the figures it looks like t goes up to 0.2 not 0.02?

We mostly focus our attention on values up to 0.02, but in some cases we do present results for a wider range of values for completeness. We now clarify this in the Methods section.

- L108ff: I found it rather hard to remember the main findings from the results. I would suggest rewriting this a little bit and adding a summarizing sentence at the end of each section

OK

- L220-221: Write out full species names

Done

- L229: Lacks references. Can you provide some examples?

Done

- Figures need higher resolution

Referee: 3

Comments to the Author(s)

The authors investigate the effect of a novel, beneficial mitotype on the distribution of maternally-inherited symbionts. They do this for three different possible symbiont types: symbionts that have a constant beneficial effect on their host, symbionts with a negative frequency-dependent benefit,

and symbionts with a positive frequency-dependent benefit via cytoplasmic incompatibility. The authors find that when the mitotype arises in an individual infected with the symbiont, the new mitotype nearly always goes to fixation. However, when the new mitotype arises in an uninfected individual, a wider variety of outcomes are possible. The new mitotype can drive the symbiont out of the population, or the new mitotype itself can be lost. Occasionally, there is also an equilibrium between hosts with the old mitotype (some or all of which are infected) and uninfected hosts with the new mitotype. The exact regions of parameter space that cause each outcome to occur depend strongly on the type of benefit the symbiont confers on its host.

This paper investigates a likely common influence on symbiont population dynamics that hasn't been modeled before. The authors' investigation of the effects on multiple types of symbionts is likely to make this interesting to a wide variety of readers: both those interested in mitochondrial effects and those interested in understanding how symbiont and host traits interact to determine symbiont population dynamics. The finding that rare symbionts are more likely to be lost due to novel beneficial mitotypes is intriguing, as are the cases where the presence of the symbiont can allow the resident and novel mitotypes to coexist. The paper is also very well-written, and the results section in particular is very clear.

I just have some small comments:

For Figure 5, I am very curious about the regions that look like they are not completely blue or yellow. Is there coexistence when the symbiont cannot persist and t is slightly greater than 0? Or when $h = 0.1$ and $t \sim 0.065$, $\mu \sim 0.02$? I would love to read a bit of intuition about what is going on in those regions.

I'm also curious why the novel mitotype doesn't seem to fully invade (or invade at all?) when t is small but nonzero and the symbiont is absent from the population.

Good spot! The issue here was that the simulations were not being run long enough to reach equilibrium (fixation of the new mitotype), hence they were returning non-fixation values. We have now adjusted the duration of the simulations, and confirm that the new mitotype reaches fixation for all the parameter combinations highlighted.

The appendix is very well-written, and the Mathematica code is also very clear. The tables giving offspring produced by each parental type were particularly nice for understanding the equations. The one point where I had trouble was in understanding why certain equilibria were biologically relevant or not. Some were obvious, but there were a few of the more complicated expressions that I wasn't able to figure out. It might be helpful to add some code to the Mathematica files that sorts the equilibria into the biologically relevant and irrelevant ones.

It is hard (maybe impossible) to ask Mathematica to pick these out based on their symbolic expressions. Instead we manually examined each in turn across the range of plausible parameter values to determine which were biologically relevant (non-negative and non-imaginary).

In Figures 2 and 3, I think it might be helpful to make some of the lines dashed or dotted to distinguish them more easily. This would be especially nice for the $b = 0.02$ case in Figure 3.

DONE

In Figure 2: When I initially looked at the region where the symbiont can persist and the novel mitotype can invade, I thought those two things could happen simultaneously. The rest of the paper makes it clear that the novel mitotype drives the symbiont extinct, but it might still be worth adding a note about this to the legend.

The reviewer is correct that the symbiont is driven extinct in this region, and we have added a note to the legend to clarify this.

Figure 3 legend: I think there is a typo. It says $c = 0.05$, but in that case the symbiont would not be able to persist in the population when $b = 0.02$ or 0.04 .

The reviewer is right – it should have been $c=0.01$ – now corrected

In Figures 1 and 4, it would be nice to have a label for the dashed line.

Added

I think there are a couple typos in the equations:

Line 571: I don't get this expression when I set the equation on line 570 to 1 and solve for t .

The reviewer is right – this has been corrected

Line 603: I think there might be an extra h in the denominator. Should it be $1 - h p_T (1 - p_T (1 - \mu))$?

The reviewer is correct – have removed extraneous 'h'

Lines 606 and 607: I think the first term in the numerator (currently h) should be \sqrt{h} .

The original version is right, obtained by solving the expression:

$$p = \frac{p(1 - \mu)}{1 - hp(1 - p(1 - \mu))}.$$

If the first term in numerator was \sqrt{h} then it would have to be written as:

$$p^* = \frac{\sqrt{h} \pm \sqrt{h - 4\mu(1 - \mu)}}{2\sqrt{h}(1 - \mu)}$$

to be equivalent, but we feel the version we present is neater.

Line 189: Should $t > 1$ should be $t > 0$?

The reviewer is right – this has been corrected

Appendix B

We thank the reviewers for their time, and that they are happy with the improved manuscript.

Referee: 3

Comments to the Author(s).

The authors are right about the math that was on lines 606 and 607 (now lines 176 and 177). I think the expression on line 680 may still have a typo -- it looks to me like the whole thing should be divided by 2. Otherwise, they addressed all my comments, and I think it is a very exciting model.

The referee is correct – there is a missing $/2$ here. The error was in the manuscript not in the maths code underpinning the simulations, so aside making it correct in text we have not had to remake any figures here.